# Transductive Zero-Shot Learning with Visual Structure Constraint

**Ziyu Wan**[*1], **Dongdong Chen**[*2], **Yan Li**[3], **Xingguang Yan**[4]
**Junge Zhang**[5], **Yizhou Yu**[6], **Jing Liao**[†1]
[1] City University of Hong Kong [2] Microsoft Cloud+AI
[3] PCG, Tencent [4] Shenzhen University [5] NLPR, CASIA [6] Deepwise AI Lab

## Abstract

To recognize objects of the unseen classes, most existing Zero-Shot Learning(ZSL) methods first learn a compatible projection function between the common semantic space and the visual space based on the data of source seen classes, then directly apply it to the target unseen classes. However, in real scenarios, the data distribution between the source and target domain might not match well, thus causing the well-known **domain shift** problem. Based on the observation that visual features of test instances can be separated into different clusters, we propose a new visual structure constraint on class centers for transductive ZSL, to improve the generality of the projection function (*i.e.*alleviate the above domain shift problem). Specifically, three different strategies (symmetric Chamfer-distance, Bipartite matching distance, and Wasserstein distance) are adopted to align the projected unseen semantic centers and visual cluster centers of test instances. We also propose a new training strategy to handle the real cases where many unrelated images exist in the test dataset, which is not considered in previous methods. Experiments on many widely used datasets demonstrate that the proposed visual structure constraint can bring substantial performance gain consistently and achieve state-of-the-art results. The source code is available at https://github.com/raywzy/VSC.

## 1 Introduction

Relying on massive labeled training datasets, significant progress has been made for image recognition in the past years [12]. However, it is unrealistic to label all the object classes, thus making these supervised learning methods struggle to recognize objects which are unseen during training. By contrast, Zero-Shot Learning (ZSL) [24, 38, 40] only requires labeled images of seen classes (source domain), and are capable of recognizing images of unseen classes (target domain). The seen and unseen domains often share a common semantic space, which defines how unseen classes are semantically related to seen classes. The most popular semantic space is based on semantic attributes, where each seen or unseen class is represented by an attribute vector. Besides the semantic space, images of the source and target domains are also related and represented in a visual feature space.

To associate the semantic space and the visual space, existing methods often rely on the source domain data to learn a compatible projection function to map one space to the other, or two compatible projection functions to map both spaces into one common embedding space. During test time, to recognize an image in the target domain, semantic vectors of all unseen classes and the visual feature of this image would be projected into the embedding space using the learned function, then nearest neighbor (NN) search will be performed to find the best match class. However, due to the existence of

---

[*]Equal contribution. Email: ziyuwan2-c@my.cityu.edu.hk, cddlyf@gmail.com
[†]The corresponding author. Email: jingliao@cityu.edu.hk

the distribution difference between the source and target domains in most real scenarios, the learned projection function often suffers from the well-known **domain shift** problem.

To compensate for this domain gap, transductive zero-shot learning [9] assumes that the semantic information (*e.g.*attributes) of unseen classes and visual features of all test images are known in advance. Different ways like domain adaption [15] and label propagation [39] are well investigated to better leverage this extra information. Recently, Zhang et al.[38] find that visual features of unseen target instances can be separated into different clusters even though their labels are unknown as shown in Figure 1. By incorporating this prior as a regularization term, a better label assignment matrix can be solved with a non-convex optimization procedure. However, their method still has three main limitations: 1) This visual structure prior is not used to learn a better projection, which directly limits the upper bound of the final performance. 2) They model the ZSL problem as a less-scalable batch mode, which requires reoptimization when adding new test data. 3) Like most previous transductive ZSL methods, they have not considered the real cases where many unrelated images may exist in the test dataset and make the above prior invalid.

Considering the first problem, we model the above visual structure prior as a new constraint to learn a better projection function rather than use the pre-defined one. In this paper, we adopt the visual space as the embedding space and project the semantic space into it. To learn the projection function, we not only use the projection constraint of the source domain data as [35] but also impose the aforementioned visual structure constraint of the target domain data. Specifically, during training, we first project all the unseen semantic classes into the visual space, then consider three different strategies ("Chamfer-distance based", "Bipartite matching based" and "Wasserstein-distance based") to align the projected unseen semantic centers and the visual centers. However, due to the lack of labels of test instances in the ZSL setting, we approximate these real visual centers with some unsupervised clustering algorithms (*e.g.*K-Means). Need to note that in our method, we directly apply the learned projection function to the online-mode testing, which is more friendly to real applications when compared to the batch mode in [38].

For the third problem of real application scenarios, since many unrelated images, which belong to neither seen nor unseen classes, often exist in the target domain, using current unsupervised clustering algorithms directly on the whole test dataset will generate invalid visual centers, thus misguiding the learning of the project functions. To overcome this problem, we further propose a new training strategy which first filters out the highly unrelated images and then uses the remaining ones to impose the proposed visual constraint. To the best of our knowledge, we are the first to consider this transductive ZSL configuration with unrelated test images.

We demonstrate the effectiveness of the proposed visual structure constraint on many different widely-used datasets. Experiments show that the proposed visual structure constraint consistently brings substantial performance gain and achieves state-of-the-art results.

To summarize, our contributions are three-fold as below:

- We have proposed three different types of visual structure constraint for the projection learning of transductive ZSL to alleviate its domain shift problem.
- We introduce a new transductive ZSL configuration where many unrelated images exist in the test dataset and propose a new training strategy to make our method work for it.
- Experiments demonstrate that the proposed visual structure constraint can bring substantial performance gain consistently and achieve state-of-the-art results.

## 2   Related Work

Unlike supervised image recognition [12] which relies on large-scale human annotations and cannot generalize to unseen classes, ZSL bridges the gap between training seen classes and test unseen classes via different kinds of semantic spaces. Among them, the most popular and effective one is the attribute-based semantic space [17], which is often designed by experts. To incorporate more attributes and save human labor, the text description-based [27] and word vector-based semantic space [7] are also proposed. Though the effectiveness of the proposed structure constraint is only demonstrated with the attribute semantic space by default, it should be general to all these spaces.

To relate the visual feature of test images and semantic attribute of unseen classes, three different embedding spaces are used by existing zero-shot learning methods: the original semantic space,

the original visual space, and a learned common embedding space. Correspondingly, a projection function is learned from the visual space to the semantic space [27] or from the semantic space to the visual space [35], or learn two projection functions from semantic and visual space to the common embedding space [4, 22] respectively. Our method uses the visual space as the embedding space, because it can help to alleviate the hubness problem [26] as shown in [35]. More importantly, our structure constraint is based on the separability of the visual features of unseen classes.

Recently, to alleviate the domain shift problem, transductive approaches [9, 19] are proposed to leverage test-time unseen data in the learning stage. For example, unsupervised domain adaption is used in [15], and transductive multi-class and multi-label ZSL are proposed in [9]. Our method also belongs to transductive approaches, and the proposed visual structure constraint is inspired by [38], but we have addressed their aforementioned drawbacks and improved the performance significantly.

## 3   Method

**Problem Definition**   In ZSL setting, we have $N_s$ source labeled samples $\mathcal{D}_s \equiv \{(x_i^s, y_i^s)\}_{i=1}^{N_s}$, where $x_i^s$ is an image and $y_i^s \in \mathcal{Y}_s = \{1, \ldots, S\}$ is the corresponding label within total $S$ source classes. We are also given $N_u$ unlabeled target samples $\mathcal{D}_u \equiv \{x_i^u\}_{i=1}^{N_u}$ that are from target classes $\mathcal{Y}_u = \{S+1, \ldots, S+U\}$. According to the definition of ZSL, there is no overlap between source seen classes $\mathcal{Y}_s$ and target unseen classes $\mathcal{Y}_u$, $i.e. \mathcal{Y}_s \cap \mathcal{Y}_u = \emptyset$. But they are associated in a common semantic space, which is the knowledge bridge between the source and target domain. As explained before, we adopt semantic attribute space here, where each class $z \in \mathcal{Y}_s \cup \mathcal{Y}_u$ is represented with a pre-defined auxiliary attribute vector $a_z \in \mathcal{A}$. The goal of ZSL is to predict the label $y_i^u \in \mathcal{Y}_u$ given $x_i^u$ with no labeled training data.

Besides the semantic representations, images of the source and target domains are also represented with their corresponding features in a common visual space. To relate these two spaces, projection functions are often learned to project these two spaces into a common embedding space. Following [35], we directly use the visual space as the embedding space, in which case only one projection function is needed. The key problem then becomes how to learn a better projection function.

**Motivation**   Our method is inspired by [38], whose idea is shown in Figure 1: thanks to the powerful discriminativity of pre-trained CNN, the visual features of test images can be separated into different clusters. We denote the centers of these clusters as *real* centers. We believe that if we have a perfect projection function to project the semantic attributes to the visual space, the projected points (called *synthetic* centers) should align with *real* centers. However, due to the domain shift problem, the projection function learned on the source domain is not perfect so that the *synthetic* centers (*i.e.*VCL centers in Figure 1) will deviate from *real* centers, and then NN search among these deviated centers to assign labels will cause inferior ZSL performance. Based on the above analysis, besides source domain data, we attempt to take advantage of the existing discriminative structure of target unseen class clusters during the learning of the projection function, *i.e.*the learned projection function should also align the synthetic centers with the real ones in the target domain.

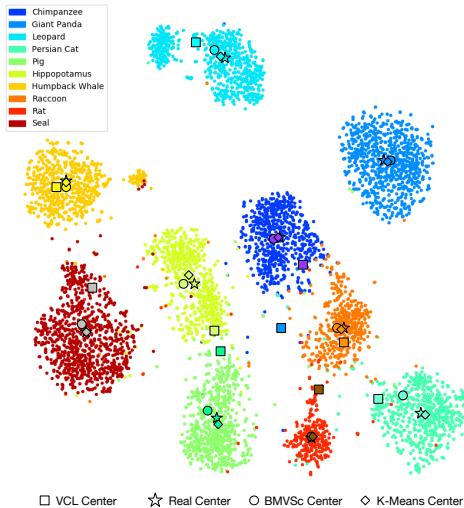

Figure 1: Visualization of CNN feature distribution of 10 target unseen classes on AwA2 dataset using t-SNE, which can be clearly clustered into several real centers (stars). Squares (VCL) are synthetic centers projected by the projection function learned only from source domain data. By incorporating our visual structure constraint, our method (BMVSc) can help to learn better projection function and the generated synthetic semantic centers would be much closer to the real visual centers.

### 3.1   Visual Center Learning (VCL)

In this section, we first introduce a baseline method which learns the projection function only with source domain data. Specifically, a CNN feature extractor $\phi(\cdot)$ is used to convert each image $x$ into a $d$-dimensional feature vector

$\phi(x) \in \mathcal{R}^{d \times 1}$. According to the above analysis, each class $i$ of source domain should have a *real* visual center $c_i^s$, which is defined as the mean of all feature vectors in the corresponding class. For the projection function, a two-layer embedding network is utilized to transfer source semantic attribute $a_i^s$ to generate corresponding synthetic center $c_i^{syn,s}$:

$$c_i^{syn,s} = \sigma_2(w_2^T \sigma_1(w_1^T a_i^s)) \tag{1}$$

where $\sigma_1(\cdot)$ and $\sigma_2(\cdot)$ denote non-linear operation (Leaky ReLU with negative slope of 0.2 by default). $w_1$ and $w_2$ are the weights of two fully connected layers to be learned.

Since the correspondence relationship is given in the source domain, we directly adopt the simple mean square loss to minimize the distance between synthetic centers $c^{syn}$ and real centers $c$ in the visual feature space:

$$\mathcal{L}_{MSE} = \frac{1}{S} \sum_{i=1}^{S} \|c_i^{syn,s} - c_i^s\|_2^2 + \lambda \Psi(w_1, w_2) \tag{2}$$

where $\Psi(\cdot)$ is the $L2$-norm parameter regularizer decreasing the model complexity, we empirically set $\lambda = 0.0005$. Need to note that different from [35] which trains with a large number of individual instances of each class $i$, we choose to utilize a single cluster center $c_i^s$ to represent each object class, and train the model with just several center points. It is based on the observation that the instances of the same category could form compact clusters, and will make our method much more computationally efficient.

When performing ZSL prediction, we first project the semantic attributes of each unseen class $i$ to its corresponding synthetic visual center $c_i^{syn,u}$ using the learned embedding network as in Equation (1). Then for a test image $x_k^u$, its classification result $i^*$ can be achieved by selecting the nearest synthetic center in the visual space. Formally,

$$i^* = \underset{i}{argmin} \ \|\phi(x_k^u) - c_i^{syn,u}\|_2 \tag{3}$$

### 3.2  Chamfer-Distance-based Visual Structure Constraint(CDVSc)

As discussed earlier, the domain shift problem will cause the target synthetic centers $c^{syn,u}$ deviated from the real centers $c^u$, thus yields poor performance. Intuitively, if we also require the projected synthetic centers to align with the real ones by using the target domain dataset during the learning process, a better projection function can be learned. However, due to the lack of the label information of the target domain, it is impossible to directly get real centers of unseen classes. Considering the fact that the visual features of unseen classes can be separated into different clusters, we try to utilize some unsupervised clustering algorithms (K-means by default) to get approximated real centers. To valid it, we plot the K-means centers in Figure 1, which are very close to the real ones.

After obtaining the cluster centers, aligning the structure of cluster centers to that of synthetic centers can be formulated as reducing the distance between the two unordered high-dimensional point sets. Inspired by the work in 3D point clouds [6], a symmetric Chamfer-distance constraint is proposed to solve the structure matching problem:

$$\mathcal{L}_{CD} = \sum_{x \in C^{syn,u}} min_{y \in C^{clu,u}} \|x - y\|_2^2 + \sum_{y \in C^{clu,u}} min_{x \in C^{syn,u}} \|x - y\|_2^2 \tag{4}$$

where $C^{clu,u}$ indicates the cluster centers of unseen classes obtained by K-means algorithm. $C^{syn,u}$ represents the synthetic target centers obtained with the learned projection. Combining the above constraint, the final loss function to train the embedding network is defined as:

$$\mathcal{L}_{CDVSc} = \mathcal{L}_{MSE} + \beta \times \mathcal{L}_{CD} \tag{5}$$

### 3.3  Bipartite-Matching-based Visual Structure Constraint(BMVSc)

CDVSc helps to preserve the structure similarity of two sets, but sometimes many-to-one matching may happen with the Chamfer-distance constraint. This conflicts with the important prior in ZSL that the obtained matching relation between synthetic and real centers should conform to the strict one-to-one principle. When undesirable many-to-one matching arises, the synthetic centers will be pulled to incorrect real centers and result in inferior performance. To address this issue, we change

CDVSc to bipartite matching based visual structure constraint (BMVSc), which aims to find a global minimum distance between the two sets meanwhile to satisfy the strict one-to-one matching principle.

We first consider a graph $G = (V, E)$ with two partitions $A$ and $B$, where $A$ is the set of all synthetic centers $Csyn, u$ and $B$ contains all cluster centers of target classes. Let $dis_{ij} \in D$ denotes the Euclidean distance between $i \in A$ and $j \in B$, element $x_{ij}$ of the assignment matrix $X$ defines the matching relationship between $i$ and $j$. To find a one-to-one minimum matching between real and synthetic centers, we could formulate it as a min-weight perfect matching problem, and optimize the problem as follows:

$$\mathcal{L}_{BM} = \min_X \sum_{i,j} dis_{ij}x_{ij}, \qquad s.t. \quad \sum_j x_{ij} = 1, \sum_i x_{ij} = 1, x_{ij} \in \{0,1\} \tag{6}$$

In this formulation, the assignment matrix $X$ strictly conforms to the one-to-one principle. To solve this linear programming problem, we employ Kuhn-Munkres algorithm whose time complexity is $O(V^2 E)$. Like **CDVSc**, we also combine the MSE loss and this bipartite matching loss

$$\mathcal{L}_{BMVSc} = \mathcal{L}_{MSE} + \beta \times \mathcal{L}_{BM} \tag{7}$$

### 3.4 Wasserstein-Distance-based Visual Structure Constraint(WDVSc)

Ideally, if the synthetic and real centers are compact and accurate, the above bipartite matching based distance can achieve a global optimal matching. However, this assumption is not always valid, especially for the approximated cluster centers of target classes, because these centers may contain noises and are not accurate enough. Therefore, instead of using a hard-value (0 or 1) assignment matrix $X$, a soft-value $X$ whose values represent the joint probability distribution between these two point sets is further considered by using the Wasserstein distance. In the optimal transport theory, Wasserstein distance is demonstrated as a good metric for measuring the distance between two discrete distributions, whose goal is to find the optimal "coupling matrix" $X$ that achieves the minimum matching distance. Its objective formulation is the same as Equation (6), but $X$ represents the soft joint probability values rather than $\{0,1\}$. In this paper, in order to make this optimization problem convex and solve it more efficiently, we adopt the entropy-regularized optimal transport problem by using the Sinkhorn iterations [5].

$$\mathcal{L}_{WD} = \min_X \sum_{i,j} dis_{ij}x_{ij} - \epsilon H(X) \tag{8}$$

where $H(X)$ is the entropy of matrix $H(X) = -\sum_{ij} x_{ij}log x_{ij}$, $\epsilon$ is the regularization coefficient to encourage smoother assignment matrix $X$. The solution $X$ can be written in form $X = diag\{u\}Kdiag\{v\}$ ($diag\{v\}$ returns a square diagonal matrix with vector $v$ as the main diagonal), and the iterations alternate between updating $u$ and $v$ is:

$$u^{(k+1)} = \frac{a}{Kv^{(k+1)}}, \qquad v^{(k+1)} = \frac{b}{K^T u^{(k+1)}} \tag{9}$$

Here, $K$ is a kernel matrix calculated with $D$. Since these iterations are solving a regularized version of the original problem, the corresponding Wasserstein distance that results is sometimes called the Sinkhorn distance. Combining this constraint, the final loss function is:

$$\mathcal{L}_{WDVSc} = \mathcal{L}_{MSE} + \beta \times \mathcal{L}_{WD} \tag{10}$$

### 3.5 A Realistic Setting with Unrelated Test Data

Existing transductive ZSL methods always assume that all the images in the test dataset belong to target unseen classes we have already defined. However, in real scenarios, many unrelated images which do not belong to any defined class may exist. If we directly perform clustering on all these unfiltered images, the approximated real centers will deviate far from the real centers of unseen classes and make the proposed visual structure constraint invalid. This problem also exists in [38]. To solve this relatively difficult setting, we propose a new training strategy for our method which first uses the baseline VCL to filter out unrelated images before conducting **CDVSc**, **BMVSc** or **WDVSc**.

**Step 1**: Since the source domain data is definitely clean, and we assume that the domain shift problem is not that severe, we first use **VCL** to get the initial unseen synthetic centers $C$.

Table 1: Quantitative comparisons of MCA (%) under standard splits (SS) in conventional ZSL setting. **I**: Inductive, **T**: Transductive, **O**: Our method, Bold: Best, Blue: Second best, V: VGG, R: ResNet, G: GoogLeNet

| | Method | Features | AwA1 | AwA2 | CUB | SUN72 | SUN10 |
|---|---|---|---|---|---|---|---|
| **I** | CONSE [24] | R | 63.6 | 67.9 | 36.7 | 44.2 | – |
| | SSE [36] | V | 76.3 | – | 30.4 | – | 82.5 |
| | JLSE [37] | V | 80.5 | – | 42.1 | – | 83.8 |
| | SynC [4] | R | 72.2 | 71.2 | 54.1 | 59.1 | – |
| | SAE [16] | R | 80.6 | 80.7 | 33.4 | 42.4 | – |
| | SCoRe [23] | V | 82.8 | – | 59.5 | – | – |
| | f-CLSWGAN [33] | R | 69.9 | – | 61.5 | 62.1 | – |
| **T** | SP-ZSR [38] | V | 92.0 | – | 53.2 | – | 86.0 |
| | DSRL [34] | V | 87.2 | – | 57.1 | – | 85.4 |
| | DMaP [19] | V+G+R | 90.5 | – | 67.7 | – | – |
| | VZSL [31] | V | 94.8 | – | 66.5 | – | 87.8 |
| | QFSL [30] | V | – | 84.1 | 61.2 | – | – |
| **O** | **VCL** | V | 81.7 | 82.6 | 58.2 | 58.8 | 87.2 |
| | **CDVSc** | V | 89.6 | 93.3 | 69.9 | 59.7 | 90.6 |
| | **BMVSc** | V | 92.7 | 94.0 | 70.8 | 61.3 | 89.7 |
| | **WDVSc** | V | 92.9 | 94.2 | 71.0 | 62.3 | 91.2 |
| | **VCL** | R | 82.0 | 82.5 | 60.1 | 63.8 | 89.6 |
| | **CDVSc** | R | 94.3 | 93.9 | **74.2** | 64.5 | 90.5 |
| | **BMVSc** | R | <span style="color:blue">95.9</span> | **96.8** | <span style="color:blue">73.6</span> | <span style="color:blue">66.2</span> | <span style="color:blue">91.7</span> |
| | **WDVSc** | R | **96.2** | <span style="color:blue">96.7</span> | **74.2** | **67.8** | **92.2** |

**Step 2**: Find distance set $\mathcal{D}$ of the farthest point pair of each source class in visual feature space.

**Step 3**: Select reliable image $x$ if and only if $\exists c_i \in C, \|x - c_i\|_2^2 \leq max(\mathcal{D})/2$ to construct a new target domain and perform unsupervised clustering on this domain.

**Step 4**: Conduct **CDVSc**, **BMVSc** or **WDVSc** as above.

## 4   Experiments

**Implementation Details**   We adopt the pretrained ResNet-101 to extract visual features unless specified. All images are resized to $224 \times 224$ without any data augmentation, and the dimension of extracted features is 2048. The hidden unit numbers of the two FC layers in the embedding network are both 2048. Both visual features and semantic attributes are L2-normalized. Using Adam optimizer, our method is trained for 5000 epochs with a fixed learning rate of 0.0001. The weight $\beta$ in CDVSc and BMVSc is cross-validated in $[10^{-4}, 10^{-3}]$ and $[10^{-5}, 10^{-4}]$ respectively, while WDVSc directly sets $\beta = 0.001$ because of its very stable performance.

**Datasets**   To demonstrate the effectiveness of our method, extensive experiments are conducted on four widely-used ZSL benchmark datasets, *i.e.*, AwA1, AwA2, CUB ,SUN10, and SUN72. Following the same configuration of previous methods, two different data split strategies are adopted: 1) **Standard Splits (SS)**: The standard seen/unseen class split is first proposed in [17] and then widely used in most ZSL works. 2) **Proposed Splits (PS)**: This split way is proposed by[32] to remove the overlapped ImageNet-1K classes from target domain since it is used to pre-train the CNN model. Please refer to the supplementary material for more details.

**Evaluation Metrics**   For fair comparison and completeness, we consider two different ZSL settings: 1) **Conventional ZSL**, which assumes all the test instances only belong to target unseen classes. 2) **Generalized ZSL**, where test instances are from both seen and unseen classes, which is a more realistic setting in real applications. For the former setting, we compute the multi-way classification accuracy (MCA) as in previous works, while for the latter one, we define three metrics. 1) $acc_{\mathcal{Y}_s}$ – the accuracy of classifying the data samples from the seen classes to all the classes (both seen and unseen); 2) $acc_{\mathcal{Y}_u}$ – the accuracy of classifying the data samples from the unseen classes to all the classes; 3) H – the harmonic mean of $acc_{\mathcal{Y}_s}$ and $acc_{\mathcal{Y}_u}$.

Table 2: Quantitative comparisons under the proposed splits (PS).

| Method | AwA2 | CUB | SUN72 | Ave. |
|---|---|---|---|---|
| CONSE [24] | 44.5 | 34.3 | 38.8 | 39.2 |
| DeViSE [7] | 59.7 | 52.0 | 56.5 | 56.0 |
| SJE[2] | 61.9 | 53.9 | 53.7 | 56.5 |
| SynC [4] | 46.6 | 55.6 | 56.3 | 52.8 |
| SAE [16] | 54.1 | 33.3 | 40.3 | 42.5 |
| SCoRe [23] | 69.5 | 61.0 | 51.7 | 60.7 |
| LDF[20] | – | 69.2 | – | – |
| PSR-ZSL[3] | 63.8 | 56.0 | 61.4 | 60.4 |
| DCN [21] | – | 56.2 | 61.8 | – |
| **VCL** | 61.5 | 59.6 | 59.4 | 60.1 |
| **CDVSc** | 78.2 | 71.7 | 61.2 | 70.3 |
| **BMVSc** | 81.7 | 71.0 | 62.2 | 71.6 |
| **WDVSc** | **87.3** | **73.4** | **63.4** | **74.7** |

Table 3: Quantitative comparisons under generalized ZSL setting.

| | AwA2 | | | CUB | | |
|---|---|---|---|---|---|---|
| Method | $acc_{\mathcal{Y}_u}$ | $acc_{\mathcal{Y}_s}$ | H | $acc_{\mathcal{Y}_u}$ | $acc_{\mathcal{Y}_s}$ | H |
| CONSE [24] | 0.5 | **90.6** | 1.0 | 1.6 | 72.2 | 3.1 |
| SSE [36] | 8.1 | 82.5 | 14.8 | 8.5 | 46.9 | 14.4 |
| DeViSE [7] | 17.1 | 74.7 | 27.8 | 23.8 | 53.0 | 32.8 |
| SJE[2] | 8.0 | 73.9 | 14.4 | 23.5 | 59.2 | 33.6 |
| ESZSL[28] | 5.9 | 77.8 | 11.0 | 12.6 | 63.8 | 21.0 |
| SynC [4] | 10.0 | 90.5 | 18.0 | 11.5 | 70.9 | 19.8 |
| ALE[1] | 14.0 | 81.8 | 23.9 | 23.7 | 62.8 | 34.4 |
| PSR-ZSL[3] | 20.7 | 73.8 | 32.3 | 24.6 | 54.3 | 33.9 |
| **VCL** | 21.4 | 89.6 | 34.6 | 15.6 | **86.3** | 26.5 |
| **CDVSc** | 66.9 | 88.1 | 76.0 | 37.0 | 84.6 | 51.4 |
| **BMVSc** | 71.9 | 88.2 | 79.2 | 33.1 | 86.1 | 47.9 |
| **WDVSc** | 76.4 | 88.1 | **81.8** | **43.3** | 85.4 | **57.5** |

## 4.1 Conventional ZSL Results

To show the effectiveness of the proposed visual structure constraint, we first compare our method with existing state-of-the-art methods in the conventional setting. Table 1 is the comparison results under standard splits (**SS**), where we also re-implement our method using 4096-dimensional VGG features to guarantee fairness. Obviously, with the three different types of visual structure constraint, our method can obtain substantial performance gains consistently on all the datasets and outperforms previous state-of-the-art methods. The only exception is that VZSL [31] is slightly better than our method on the AwA1 dataset when using VGG features.

Specially, comparing with SP-ZSR [38] which shares the similar spirit with our method, we could find that their performance sometimes is even worse than inductive methods such as SynC [4], SCoRe [23] or VCL. The possible underlying reason is that, when utilizing the structure information only in test time, the final performance gain highly depends on the quality of the project function. When the projection function is not good enough, the initial synthetic centers will deviate far from the real centers and result in bad matching results with unsupervised cluster centers, thus causing even worse results. By contrast, in our method, this visual structure constraint is incorporated into the learning of projection function in the training stage, which can help to learn a better projection function and bring performance gain consistently. Another bonus is that, during runtime, we can directly do recognition in real-time online-mode rather than the batch-mode optimization in SP-ZSR [38], which is more friendly in real applications.

The results on proposed splits of AwA2, CUB and SUN72 are reported in Table 2 with ResNet-101 features. It can be seen that almost all methods suffer from performance degradation under this setting. However, our proposed method could still maintain the highest accuracy. Specifically, the improvements obtained by our method range from 0.8% to 25.8%, which indicate that visual structure constraint is effective to solve the domain shift problem.

## 4.2 Generalized ZSL Results

In Table 3, we compare our method with eight different generalized ZSL methods. It can be seen that, although almost all the methods cannot maintain the same level accuracy for both seen ($acc_{\mathcal{Y}_s}$) and unseen classes ($acc_{\mathcal{Y}_u}$), our method with visual structure constraint still significantly outperforms other methods by a large margin on these datasets. More specifically, take CONSE [24] as an example, due to the domain shift problem, it can achieve the best results on the source seen classes but totally fails on the target unseen classes. By contrast, since the proposed two structure constraints can help to align the structure of synthetic centers to that of real unseen centers, our method can achieve acceptable ZSL performance on target unseen classes.

## 4.3 Results of New Realistic Setting

To imitate the realistic setting where many unrelated images may exist in the test dataset, we mix the test dataset with extra 8K unrelated images from the aPY dataset. These unrelated images do not belong to the classes of either AwA2 or ImageNet-1K. From Table 4, it could be seen that without

Table 4: Results (%) on more realistic setting. With the new proposed training strategy ($S + *$), the proposed method can still work well and bring performance gain.

| Method | VCL | CDVSc | BMVSc | WDVSc | $S$+CDVSc | $S$+BMVSc | $S$+WDVSc |
|---|---|---|---|---|---|---|---|
| SS+noise | 82.5 | 79.7 | 78.3 | 81.3 | 89.3 | 86.9 | **92.4** |
| PS+noise | 61.5 | 57.4 | 58.9 | 60.8 | 65.3 | 66.7 | **78.3** |

Table 5: Generality to the word vector based semantic space on the AwA1 dataset.

| Method | DeViSE[7] | ZSCNN[18] | SS-Voc[10] | DEM[35] | VCL | CDVSc | BMVSc | WDVSc |
|---|---|---|---|---|---|---|---|---|
| **MCA (%)** | 50.4 | 58.7 | 68.9 | 78.8 | 72.3 | 79.4 | 83.9 | **90.8** |

filtering out the unrelated images, the performance of our method with CDVSc, BMVSc and WDVSc degrades, which means that the alignment of wrong visual structures is harmful to the learning of projection function. By contrast, the new training strategy ($S + *$) can still make the proposed visual structure constraint work very well.

## 4.4 More Analysis

Due to the limited space, only two analysis experiments are given in this section. Please refer to the supplementary material for more analysis.

**Generality to word vectors based semantic space.** Compared to some previous methods which are only applicable to one specific semantic space, we further demonstrate that the proposed visual structure constraint can also be applied to word vector-based semantic space. Specifically, to obtain the word representations for the embedding networks inputs, we use the GloVe text model [25] trained on the Wikipedia dataset leading to 300-d vectors. For the classes containing multiple words, we match all the words in the trained model and average their word embeddings as the corresponding category embedding. As shown in Table 5, though the contained effective information of word vectors is less than that of semantic attributes, the proposed visual structure constraint can still bring substantial performance gain and outperform previous methods. Note that DEM [35] utilized 1000-d word vectors provided by [8, 9] to represent a category.

**Robustness for imperfect separability of visual features for unseen classes.** Though our motivation is inspired by the great separable ability of visual features for unseen classes on the AwA2 dataset, we find the proposed visual structure constraint is very robust and does not rely on it seriously. For example, on the CUB dataset, the feature distribution (refer to Fig 5 in the supplementary material) is not totally separable, but the proposed visual structure constraint still brings significant performance gain as shown in the above Tables. Because even though there are some incorrect clusters, as long as most of them are correct clusters, the proposed visual structure constraint will be beneficial.

On the other hand, though the unseen class number $K$ is often pre-defined, we find the proposed visual constraint can improve the performance even when it is unknown. In Table 6, we report the performance for different $K$. Specifically, we first perform K-means both in the semantic space and visual space simultaneously, then use BMVSc and WDVSc to align these two synthetic sets. Obviously, the proposed visual structure constraint can bring performance gain consistently. With the increase of K, it could capture the more fine-grained structure of visual space and achieve better results. In other words, as long as the visual features can form some superclasses (not fine-level, which is satisfied by most datasets), the proposed visual structure constraint is always effective.

## 5 Conclusion

To alleviate the domain shift problem in ZSL, three new different types of visual structure constraint are proposed for transductive ZSL in this paper. We also introduce a new transductive ZSL configuration for real applications and design a new training strategy to make our method work well. Experiments demonstrate that they can bring substantial performance gain consistently on different benchmark datasets and outperform previous state-of-the-art methods by a large margin. In the future, we will try to apply the proposed idea to broader application scenarios [13, 11, 14, 29].

Table 6: Results (%) of different cluster number $K$ on AwA2 dataset.

| K | 0 | 3 | 4 | 5 | 6 | 7 | 8 | 9 | 10 |
|---|---|---|---|---|---|---|---|---|----|
| **BMVSc** | 61.5 | 62.3 | 62.9 | 64.5 | 65.1 | 68.2 | 70.1 | 74.3 | 81.7 |
| **WDVSc** | 61.5 | 63.4 | 64.0 | 66.3 | 67.0 | 69.2 | 75.1 | 80.3 | 87.3 |

# 6 Acknowledgements

We would like to thank the anonymous reviewers for their thoughtful comments and efforts towards improving our work. This work was supported by the Natural Science Foundation of China (NSFC) No.61876181, CityU start-up grant 7200607 and Hong Kong ECS grant 21209119.

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
