[Supplementary Material · NIPS_Supple.pdf]

# Supplementary Material - Transductive Zero-Shot Learning with Visual Structure Constraint

**Ziyu Wan**[*1], **Dongdong Chen**[*2], **Yan Li**[3], **Xingguang Yan**[4]
**Junge Zhang**[5], **Yizhou Yu**[6], **Jing Liao**[†1]
[1] City University of Hong Kong [2] Microsoft Cloud+AI
[3] PCG, Tencent [4] Shenzhen University [5] NLPR, CASIA [6] Deepwise AI Lab

In this document, we provide additional analysis to supplement our main submission. In Sec. 1, we show the qualitative results of BMVSc on AwA2 and CUB. In Sec. 2, the experiment details of main paper are given. In Sec. 3, we provide more analysis which is not put in the main paper due to the limited space. In Sec. 4, we explain the reason of choosing visual space as embedding space.

## 1 Qualitative Results

In Figure 1, we have shown some qualitative results of the proposed **BMVSc** on the AwA2 and CUB datasets. Although the test images of each class have an overall different appearance, the projection function learned by our method can still capture important discriminative semantic information from their visual characteristics, which corresponds to their semantic attributes. For example, the predicted sheep images in AwA2 all share *furry*, *bulbous* and *hooves* attributes. However, we could also observe some misclassified images such as the walrus in row 6 of AwA2. After careful analysis, we find two possible reasons: 1) The discriminative ability of the pretrained CNN is not enough to separate the visual appearances between too similar categories. In fact, the visual appearance of seal and walrus are so close that even people could not distinguish them by rule and line without expert knowledge. This problem can be solved only by more powerful visual features. 2) Some attribute annotations are not accurate enough. For example, the seal category possesses *spots* of semantic descriptions, but walrus does not, but both these two categories own this attribute in the semantic annotation. Such incorrect supervision information will mislead the learning of the projection function.

## 2 Experiment Details

**Datasets** Extensive experiments are conducted on three widely-used ZSL benchmark datasets, , Animals with Attributes2 (AwA2) [8], Caltech-UCSD Birds 200-2011 (CUB) [7] and Scene UNderstanding (SUN) [5]. The statistics of these datasets are briefly introduced as below:

- **Animals with Attributes1 (AwA1)** [2] For fair comparison with previous methods, we report the results on **AwA1** which is an old version of animal datasets of ZSL. There are totally 30,475 images coming from 50 different classes in AwA1, and 85-dim continuous attributes are employed as semantice space.

- **Animals with Attributes2 (AwA2)** [8] contains 37,322 images from 50 animals categories, where 40 of 50 classes are used for training and the rest 10 are used for testing. We adopt the class-level continuous 85-dim attributes as the semantic representations.

- **Caltech-UCSD Birds 200-2011 (CUB)** [7] is a fine-grained bird dataset with 200 species of birds and 11,788 images. Each image is associated with a 312-dim continuous attribute vector. Following [8], we use the class-level attribute vector and the 150/50 split.

---

[*]Equal contribution.
[†]The corresponding author.

Figure 1: Qualitative results of BMVSc on 6 categories of AwA2 and CUB datasets. We list the top-6 images classified to each category. The misclassified images are marked with red bounding boxes and the right name of category is below the corresponding image.

- **SUN-Attribute (SUN)** [5] includes 14,340 images coming from 717 scenes. Each sample is paired with a binary 102-dim semantic vector. We compute class-level continuous attributes as our semantic representations by averaging the image-level attributes for each class. 707/10 (**SUN10**) and 645/72 (**SUN72**) splits are adopted in our experiments.

**Evaluation Metrics**  Following previous work [9], the multi-way classification accuracy is adopted as our evaluation metric:

$$acc_{\mathcal{Y}} = \frac{1}{\|\mathcal{Y}\|} \sum_{i=1}^{\|\mathcal{Y}\|} \frac{\#\ \text{correct predictions in i}}{\#\ \text{samples in i}} \tag{1}$$

We also adopt the same data splits as [9] and denote $acc_{\mathcal{Y}_s}$ and $acc_{\mathcal{Y}_u}$ as the accuracy of images from the seen and unseen classes respectively. Moreover, the harmonic mean is computed to measure the ZSL performance in the generalized setting with the same weights of $acc_{\mathcal{Y}_s}$ and $acc_{\mathcal{Y}_u}$:

$$H = \frac{2 * acc_{\mathcal{Y}_u} * acc_{\mathcal{Y}_s}}{acc_{\mathcal{Y}_u} + acc_{\mathcal{Y}_s}} \tag{2}$$

Figure 2: The comparison of convergence curve between instance-based method (DEM) and our center-based method (VCL).

Figure 3: Matching matrixs between the projected semantic centers and visual cluster centers of **CDVSc** (left) and **BMVSc** (right) on the AwA2 dataset. **BMVSc** can guarantee strict one-one matching while **CDVSc** may have many-to-one matching.

Figure 4: The right matching number and distance between the projected semantic centers and real visual centers during the training of BMVSc on the SUN dataset.

## 3   More analysis

**Possible many-to-one problem in CDVSc.**   To verify that there may exist many-to-one matching problem during the training of **CDVSc**, we randomly select the output of embedding networks of one epoch and visualize the matching results on the AwA2 dataset in Figure 3. It can be seen that one projected semantic center can be matched by multiple visual cluster centers, and vice versa. By contrast, **BMVSc** can guarantee strict one-one matching, which may be the reason of better results shown in main text on this dataset.

**Progressive improving of center matching in BMVSc.**   The final ZSL performance depends on the alignment of the projected semantic centers and real visual centers. In our method we use K-means cluster centers to approximate the real centers and minimize their matching distance. So one natural question would be "whether we can achieve this final objective by training with cluster centers from K-means?". To answer this question, we calculate the the number of right matching point and distances between the projected semantic centers and real visual centers respectively during the training of **BMVSc** . We plot these two metrics of the SUN dataset in Figure 4. Obviously, **BMVSc** can definitely improve the matching of the projected semantic centers and real visual centers by only using the cluster centers from K-means.

**Center-based objective *vs* Instance-based objective**   Compared to previous instance-based optimization objective, our center-based optimization objective is much more computationally efficient. To verify this point, we re-implement the work **DEM** [10] and adopt the same network structure, parameter setting and optimization algorithm with our VCL method on the AwA2 dataset. Then we plot the change of loss and accuracy with epoch increasing in Figure 2 respectively. It shows that our center-based optimization objective converges faster than previous instance-based optimization objective and can even achieve slightly better final results.

**Why slightly worse results are obtained by BMVSc than CDVSc on the CUB dataset?**   In our paper, three different types of visual structure constraint are proposed to alleviate the domain shift problem in ZSL. BMVSc can solve the possible many-to-one matching problem in CDVSc and satisfy the strict one-to-one principle, which potentially helps to achieve better results, such as the gain can be observed on the AwA2 and SUN datasets. However, on the CUB dataset, the performance of BMVSc is slightly worse than CDVSc. Although this difference is quite subtle when it is compared to the absolute gain coming from the visual structure constraint, we still want to find the possible underlying reason.

To answer this question, we first plot the feature distribution of all the categories in the left subfigure in Figure 5 with TSNE. We could find that the feature distribution of some categories is too close to be distinguished because the feature pretrained on ImageNet is not representative enough for this CUB dataset. This somehow violates our assumption that the separated clusters of unseen classes obtained from pre-trained CNN models are already discriminative, and thus lead to this degradation phenomenon. To verify it, we check the matching matrix obtained by our methods and find that there indeed exists wrong matches due to very closed real centers. Specifically, consider synthetic center

(a) Visualization of ResNet-101 features of 50 classes on the CUB dataset.

(b) Matching relations between synthetic center and two similar real centers. Red line denotes BMVSc matching, and green line and red line denote CDVSc matching.

Figure 5: Analysis to find the possible underlying reason why slightly worse performance is observed by BMVSc on the CUB dataset.

$X$ of *yellow billed cuckoo*, and two similar real centers $Y$ and $Z$ of *mangrove cuckoo* and *yellow billed cuckoo*. $X - Z$ is the right matching, and $X - Y$ is the wrong matching. In BMVSc, if the wrong matching happens, $X$ will be pulled closer to inaccurate center $Y$ (loss term: $\|X - Y\|_2^2$). By contrast, the contribution of CDVSc to the final loss is $\frac{\|X-Y\|_2^2 + \|X-Z\|_2^2}{2}$, which will also force $X$ to approach $Z$ and alleviate the wrong matching problem to some certain degree.

**Importance of unsupervised cluster centers and semantic attributes.** In our method, to recognize the target domain images, two different types of knowledge are leveraged: unsupervised cluster centers and semantic attributes. To study the importance of these two components, we design a simple voting algorithm to calculate the upper bound of unsupervised clustering algorithms for ZSL recognition. Specifically, we assume the ground truth label for each unseen instance is accessible. Then for each cluster center obtained by K-means, we predict its category through a voting process, its category is the one which most images in this cluster belong to. Finally, the classification results for test instances are directly set to the label of the corresponding cluster. In this way, because we have already used the ground truth information, it can be viewed as the upper bound of K-means clustering algorithm. As shown in Table 1, its performance is even better than our baseline **VCL**. which demonstrates that the unsupervised clustering information is very useful. By combining the semantic attributes and this unsupervised cluster information during the learning process, our method **CDVSc**, **BMVSc** and **WDVSc** are all better than the upper bound of **K-Means** and **VCL**.

## 4 Hubness Problem

Hubness phenomenon means that several objects may occur as nearest neighbor of most points in a high-dimensional vector space for nearest neighbour (NN) search. [6] shows that hubness is a special property inheriting from data distribution and the key to tackle the hubness problem is to choose the right embedding space.

|          | AwA2 | CUB  | SUN  |
|----------|------|------|------|
| K-Means  | 75.0 | 67.4 | 57.6 |
| VCL      | 61.5 | 59.6 | 59.4 |
| CDVSc    | 78.2 | 71.7 | 61.2 |
| BMVSc    | 81.7 | 71.0 | 62.2 |
| WDVSc    | 87.3 | 73.4 | 63.4 |

Table 1: Analysis to demonstrate the importance of unsupervised cluster centers and semantic attributes. By combining these two types of information during training, our **CDVSc**, **BMVSc** and **WDVSc** achieve better results than the upper bound of **K-Means** and **VCL**.

Zero-Shot Learning (ZSL) aims to recognize unseen objects by learning a shared embedding space which the visual features of object instances and semantic representations of object categories can be projected to. Then in the test stages, unseen image instances will be mapped into the embedding space, where the nearest neighbour search is operated to finish classification. Following this approach, most of existing ZSL methods choose the semantic space as the embedding space, and aim to learn a projection $W$: $\mathbb{R}^n \to \mathbb{R}^m$ from visual space to semantic space. The projection function can be learned through standard ridge regression or neural networks. Formally,

$$W = \underset{W}{argmin} \|WX - Y\|^2 + \lambda \|W\|^2 \tag{3}$$

Leaning such a visual to semantic projection for ZSL has achieved promising ZSL resutls [4, 1, 3]. However, we will show that selecting the semantic space as the embedding space will make the hubness problem worse. The proof procedure contains two parts:

- Using semantic space as the embedding space will aggravate shrinkage degree of projected objects towards the origin point.
- The points closer to the origin are much more possible to become hubs.

The proof of the first part has been provided in [10], and we will give the proof of the second part in this paper.

**Definition 1** *Let $\mathcal{E}[\cdot]$ and $\mathcal{V}[\cdot]$ be the expectation and variance respectively. Let $f(\cdot)$ be the squared norm of vectors.*

**Definition 2** *Let $\mathbf{x} = [x_1, \ldots, x_n]^T$ be a random vector sampled from distribution $\mathcal{X}(0, \sigma)$. Let $\mathbf{y} = [y_1, \ldots, y_n]^T \sim \mathcal{Y}$, where $\mathbf{y}$ is also a n-dimensional vector and $\mathcal{Y}$ is a normal distribution with mean 0 and variance $d^2$. Further denote $\delta$ as the standard deviation of $\|\mathbf{y}\|^2$, i.e. $\delta = \sqrt{\mathcal{V}_\mathcal{Y}[\|\mathbf{y}\|^2]} = \sqrt{\mathcal{V}_\mathcal{Y}[f(\mathbf{y})]}$, where $\|\mathbf{y}\|^2$ could be viewed as the distance from $\mathbf{y}$ to the origin.*

**Proposition 1** *Consider $\mathbf{x}$ as a query. Assume there are two points $\mathbf{y}_1$ and $\mathbf{y}_2$, we are interested in that which vector is more likely to be closer to $\mathbf{x}$. Formally, we have*

$$f(\mathbf{y}_1) - f(\mathbf{y}_2) = \alpha\delta \tag{4}$$

*Then the expected difference $\Delta$ between the distance from $\mathbf{x}$ to $\mathbf{y}_1$ and $\mathbf{y}_2$ could be written as*

$$\Delta = \mathcal{E}_\mathcal{X}[f(\mathbf{x} - \mathbf{y_1})] - \mathcal{E}_\mathcal{X}[f(\mathbf{x} - \mathbf{y_2})] \tag{5}$$

$$\Delta = \sqrt{2n}\alpha d^2 \tag{6}$$

**Proof 1** *The expectation of distance from $\mathbf{x}$ to $\mathbf{y}_i (i = 1, 2)$ is*

$$\begin{aligned} \mathcal{E}_\mathcal{X}[f(\mathbf{x} - \mathbf{y}_i)] &= \mathcal{E}_\mathcal{X}[f(\mathbf{x})] + \mathcal{E}_\mathcal{X}[f(\mathbf{y}_i)] - 2\mathcal{E}_\mathcal{X}[\mathbf{x}]^T \mathcal{E}_\mathcal{X}[\mathbf{y}_i] \\ &= \mathcal{E}_\mathcal{X}[f(\mathbf{x})] + f(\mathbf{y}_i) \end{aligned} \tag{7}$$

*since $\mathcal{E}_\mathcal{X}[\mathbf{x}]$ equals zero. Through Equation 4, 5 and 7 we have*

$$\begin{aligned}
\Delta &= \{\mathcal{E}_\mathcal{X}[f(\mathbf{x})] + f(\mathbf{y}_1)\} - \{\mathcal{E}_\mathcal{X}[f(\mathbf{x})] + f(\mathbf{y}_2)\} \\
&= f(\mathbf{y_1}) - f(\mathbf{y_2}) \\
&= \alpha\delta
\end{aligned} \tag{8}$$

*Let $\mathbf{z}$ be a n-dimensional vector sampled from multivariate normal distribution $\mathcal{N}(0, I)$, we could get $\mathbf{y} = d\mathbf{z}$. It could be noted that actually $f(\mathbf{z})$ is distributed according to the chi-squared distribution with n degrees of freedom, i.e. $f(\mathbf{z}) \sim \chi^2(n)$, and its mean and variance are n and 2n respectively. Hence,*

$$\delta = \sqrt{\mathcal{V}_\mathcal{Y}[f(\mathbf{y})]} = \sqrt{\mathcal{V}_\mathcal{Z}[d^2 f(\mathbf{z})]} = d^2 \sqrt{\mathcal{V}_\mathcal{Z}[f(\mathbf{z})]} = d^2 \sqrt{2n} \tag{9}$$

*Substituting this result in Equation 8, we could obtain Equation 6 finally.*

In Equation 6, $\Delta$ increases with $\alpha$ meaning $\mathbf{y}_2$ would be closer to $\mathbf{x}$ than $\mathbf{y}_1$. Meanwhile, due to the growth of $\alpha$, $\mathbf{y}_2$ will shift to the origin gradually through Equation 4. Hence, it could be illustrated that points which are closer to original point tent to be hubs.

Through these two parts of proof, we could conclude that utilizing shared semantic embedding space would make hubness worse.

Our proposed method could mitigate the shrinking degree to origin points with reversing projection direction and utilizing visual space as embedding space, so it is helpful to alleviate hubness problem.