[Reviews · NeurIPS 2019]

Reviewer 1



Originality: The three ways proposed in the paper to deal with domain shift in the transductive setting is new as far as I know (although I have not got a deep knowledge of the literature). Quality: The paper is mainly an empirical one, and as such the claims are supported by the results reported in the experimental section. Clarity: The paper reads well in general, although some parts can be improved: in particular I did not understand step 3 of Sec. 3.5, what does it mean to select a reliable image? and why a new target domain?. Also the introduction is too focused on comparing this work with that of [34], while barely describing that paper. I have the impression that in order to understand all this the reader needs to go through that paper. Significance: results reported are state of the art in common ZSL benchmarks, and the implementation of the approach is likely doable by external researchers by following descriptions of the paper. Regarding table 3: why results are not reported on the SUN dataset like in the other tables? Also the approach in [29] reports better results on CUB accuracy on unseen (43.7), it would be fair to add it.

Reviewer 2



Strength: - The paper proposes an interesting and novel approach for transductive zero-shot learning. - The paper extensively evaluates their approach on three datasets under the normal and generalized Zero-shot learning setting showing strong, improving in most cases SOTA performance. - The paper proposes a new evaluation setting which include distractors from unknown classes. - Extensive supplemental material with additional analysis and details. - Overall clearly written and easy to follow. Weaknesses: 1. Experimental evaluation 1.1. It would be great to also include zero shot performance on ImageNet (this is most likely missing as there are not attribute annotations for ImageNet, but the approach does not seem to be limited to attributes for transfer) 1.2. It would be interesting to quantitatively compare to [31] and [34] as ablations of the author’s appraoch from which authors took inspiration. 1.3. The authors claim in the reproducibility checklist to have “Clearly defined error bars” and “A description of results with central tendency (e.g. mean) & variation (e.g. stddev)”, but they don’t, although it would be good if they had. 2. Related work 2.1. The paper misses to discuss (qualitatively and quantitatively) recent related work including [A]. [A] achieves higher performance on SUN. Similarly, DCN [18] is not discussed and misses in Table 3, although it is better than other prior work on CUB. 2.2. Not w.r.t. performance, but with w.r.t. transductive label propagation, [B] is relevant. 3. Clarity: 3.1. Calling splits “Proposed Splits“, although they have been proposed in [28] is a bit confusing. Better might be to refer e.g. as “pure” as in [B]. Summary and Conclusion: While the paper includes an interesting novel transductive approach for zero shot learning, with an overall solid evaluation setup and several datasets, the paper misses to compare (discussion and quantitative) to Neurips 2018 papers ([A], [18]). References: [A] Zhao, An, et al. "Domain-invariant projection learning for zero-shot recognition." Advances in Neural Information Processing Systems. 2018. [B] Transfer Learning in a Transductive Setting; Rohrbach et al. Neurips 2013 == Post rebuttal == The authors provided additional convincing results on ImageNet (and Sun) and promising to add missing comparison in the final version. Furthermore, the authors included more comparisons to prior work in the author response. While I think this paper is sufficiently different with [34], I think the discussion on [34] could be further improved. I agree with R3 that the "new setting" is somewhat adhoc, and it would probably be good to compare it to the open world setting but it is still interesting that this paper studies it, especially in the transductive setting. I expect the authors include the results from the author response and I strongly recommend that the author release code (which is missing for [34]) to allow future work to build on and compare to this work. Overall, I think the paper provides sufficient methodological and significant experimental contribution, including reporting Generalized ZSL Results. I recommend accepting this paper.

Reviewer 3



The paper proposes a method for transductive zero-shot learning. It modifies nearest center classifiers on the unseen classes by aligning with clustering “prior” based on K-means. Three approaches are proposed for aligning/matching the centers (see $3.2, $3.3, $3.4). Strengths - Strong empirical results - Simple and sound approach Weaknesses 1. Novelty/Significance/Soundness 1.1 The idea of clustering unlabeled unseen classes (transductive setting) was explored in [34, A] (Please discuss the difference with A.) The discussion on the differences between this work and [34] in L42-46 does not suggest that the proposed idea is more than incremental, especially on 2) and 3) which are about optimization details and an extension of transductive setting rather than about the method itself. 1) is also too ambiguous/not precise enough to allow me to quantify its significance. In my opinion, it would be better to describe [34] in more detail and explicitly point out why the proposed formulation is much better than what was proposed there. [A] Shojaee and Baghshah. Semi-supervised Zero-Shot Learning by a Clustering-based Approach. 2016. 1.2 The proposed approach is grounded in the existence of discriminative clusters. This is shown in Fig. 1 but on other datasets especially on find-grained datasets or where the number of unseen classes is large, this assumption could break or will not operate as well. If possible, the authors could show something similar to Fig. 1 for all datasets. Another drawback of the proposed approach is that using nearest center classifiers makes knowing of the number of clusters very important (In this paper, it is assumed to be the number of unseen classes; please make this clear in $3.2). Table 4 supports this fact and this is on the easy dataset AwA2 in which there are only 10 unseen classes. 1.3 The proposed extension of “transductive” setting ($3.5) seems adhoc and there is not really any evaluation to support that it works well. 2. Experiments Experiments can be geared more toward showing that the domain shift problem has been resolved. Can we use quantitative measures / intrinsic evaluation of centers, before and after matching, to showcase this? In $4.2, the “vanilla” harmonic mean is problematic. See Sect. 4.4.2 and especially Fig. 4 in [C] and [B]. This makes the discussion in L277-279 kind of invalid. I would also encourage evaluation using AUSUC. [B] Le Cacheux et al. From Classical to Generalized Zero-Shot Learning: A Simple Adaptation Process. 2019. [C] Changpinyo et al. Classifier and Exemplar Synthesis for Zero-Shot Learning. IJCV, 2019. 3. Discussion of related work should be improved. Besides discussion with respect to [34] which is my main concern above, more credit should be given to zero-shot learning approaches that take advantage of clustering structures by using nearest center classifiers for zero-shot learning. Besides [31] which is mentioned in this paper, see [C, D, E]. [D] Mensink et al. Distance-based image classification: Generalizing to new classes at near-zero cost. TPAMI 2013. [E] Changpinyo et al. Predicting Visual Exemplars of Unseen Classes for Zero-Shot Learning. ICCV 2017. Note that L150-154 is precisely what [E] mentions. Please also cite related work regarding the domain shift problem as well as generalized zero-shot learning. Please fix L237; Splits on all datasets do not belong to [14]. See Table 1 in [C]. 4. The paper is recommended to be revised for its English usage. Examples are a few instances of “Need to note that”, “the real cases” (L45), “valid” (L167), “re-implement” -> “test”? (L252). ### Updated Reviews ### Overall, several of my concerns are resolved and experiments do look stronger. However, I still have a strong concern regarding the relationship to [34] (see below) and in general discussion of related work. Given my current understanding of [34] and related work, the degree of the method's significance in terms of novelty, simplicity, or technical depth is in question, making it hard for me to be on the acceptance side. Taking the rebuttal and other reviews into account, I am happy to increase my score to 5. ***Key differences with [34]*** IMO, the paper did not make it clear about the relationship of this work to [34]. After reading the rebuttal, I was still unsure; thus, I checked [34] myself. With my limited understanding of what [34] did, I still have the following questions/concerns. First, the statement in the rebuttal “[34] is to improve label assignment over naive NN using a fixed project function while we aim to learn better projection function by only using naive NN assignment” doesn’t add much to L47-58. In particular, I do not see a clear difference; why should we consider the process of reassigning cluster labels (and thus the centers) in [34] as simply improving label assignment but not adapting the projection function? Second, [34] builds an adaptation/transduction technique on top of JSLE [33] which is a much weaker baseline than VCL (see the comparison in Table 1, especially on CUB and SUN10 --- the datasets where see larger improvement of the proposed three variants over [34]). In other words, how much of the gain is simply due to the focus on nearest center classifier in the visual feature space? Third, the rebuttal mentions that one of the variants WDVSc is different because it uses soft matching vs. hard matching. This point is valid but leaves the question of whether what would happen if the method in [34] uses soft matching. Please also note that most of the baselines, both in the main text and the rebuttal, are inductive zero-shot learning methods. In particular, the only comparison we have to transductive ZSL baselines ([34] and 4 others) are only presented in Table 1. To be fair, this is in part due to the fact that the new proposed splits in and the use of ResNet [28] were proposed in 2017 and not yet adopted by the transductive ZSL community. ***Key differences with [A]*** I am good with not comparing with the unpublished [A] in detail but I think the paper can still discuss it. ***Dependence on discriminative clusters and known cluster number K*** This concern in my review 1.2 is to point out that this is where the proposed method could break down. I reread L303 -317 and am OK with the argument that the predicted cluster labels could still be useful even though the features may not be so discriminative. Moreover, ImageNet results the rebuttal help resolve the concern to some degree regarding the number K. However, it would be great to discuss why the method is less brittle on ImageNet, in stark contrast to the results in Table 6. It would also be nice to have results when K > the number of unseen classes and in the more adversarial setting proposed in this paper ($3.5). Finally, I do not know the details on these experiments due to the space limit on the rebuttal, but it would be nice to know why VCL is much worse than EXEM for Hit@K >=10 even though the two methods are very similar. ***New setting*** My opinion toward the new setting both in terms of the approach ($3.5) and results ($4.3) remain the same. The approach looks quite straightforward/adhoc and the results look preliminary, so it is hard for me to count this as a significant contribution. This is also orthogonal to the main contribution of the paper. ***AUSUC and domain shift results in Table 2 of the rebuttal.*** I appreciate these new results.

[Author Response · NeurIPS 2019]

Table 1: **Left**: SUN gZSL results, **Middle**: ImageNet results, **Right**: Imagenet results with different unknown $K$.

| | $accy_u$ | $accy_s$ | $H$ |
|---|---|---|---|
| ALE[1] | 21.8 | 33.1 | 26.3 |
| VCL | 10.4 | 63.4 | 17.9 |
| CDVSc | 27.8 | 63.2 | 38.6 |
| BMVSc | 29.9 | 62.9 | 40.6 |
| WDVSc | **30.5** | **63.1** | **41.1** |

| Test data | Method | Hit@K (%) | | | | |
|---|---|---|---|---|---|---|
| | | 1 | 2 | 5 | 10 | 20 |
| ILSVRC2010 | **DIPL** | – | – | 31.7 | – | – |
| 2-hop | CONSE | 8.3 | 12.9 | 21.8 | 30.9 | 41.7 |
| | SYNC | 10.5 | 16.7 | 28.6 | 40.1 | 52.0 |
| | EXEM | 12.5 | 19.5 | 32.3 | 43.7 | 55.2 |
| 2-hop | VCL | 12.3 | 19.3 | 31.3 | 40.3 | 48.7 |
| | **WDVSc** | **17.6** | **26.7** | **38.8** | **47.5** | **57.9** |

Table 2: **Left**: AUSUC evaluation, **Middle**:Center distances, **Right**: SUN feature distribution (better zoom-in).

| | AwA2 | CUB | SUN |
|---|---|---|---|
| VCL | 0.47 | 0.31 | 0.20 |
| CDVSc | 0.74 | 0.55 | 0.35 |
| BMVSc | 0.76 | 0.51 | 0.37 |
| WDVSc | **0.79** | **0.61** | **0.38** |

| | AwA2 | CUB | SUN | ImageNet |
|---|---|---|---|---|
| VCL | 0.07 | 0.07 | 0.04 | 0.13 |
| CDVSc | 0.04 | 0.04 | 0.03 | – |
| BMVSc | 0.03 | 0.04 | 0.02 | – |
| WDVSc | **0.01** | **0.03** | **0.02** | **0.05** |

We thank all the reviewers for their valuable comments. We are very encouraged by the recognition of novelty and
performance boost from R1, R2. For each question from reviewers, we give strong experiment results and clarifications.
Missing references, minor errors, and rephrasing introduction (R1) will be fixed in the revision because of page limit.
**R1:Clarification of step 3 in Sec 3.5.** Because some unrelated images will make the approximated centers deviate
from the real centers. Therefore, we regard images whose distance to the original approximated centers is below one
threshold as reliable ones (denote as "a new target domain") and get new better-approximated centers based on them.
**R1:SUN results.** Due to limited space, SUN results are not given in the original submission, which is shown in the left
of Tab.1 now. Note that ALE is the best among SOTA methods on this dataset, but is still far behind our results.
**R1: Comparison with [29] in Table 3.** Will add it. And our harmonic means of gZSL on AwA2,CUB,and SUN (**76.4,**
**57.5, 41.1**) are all better than [29] (59.6, 49.7, 39.4) except a little lower unseen accuracy (43.3) on CUB.
**R2:ImageNet results.** WDVSc Results are shown in the middle of Tab.1, which outperform previous SOTA and
baseline VCL by more than 5 points. Even without knowing $K$ value (Right), our results consistently improve over the
VCL. This further demonstrates the superiority of the proposed visual constraints. Due to limited computation resources
in such a short period, the results of BMVSc and CDVSc are not reported here but will be included in the final version.
**R2:Comparison with [A,18,31,34].** In supp Fig.2, we have provided the convergence comparison with [31]. On AwA1
and CUB dataset, our result (**96.2, 74.2**) is much better than [31]'s result (86.7, 58.3). Because [34] has not reported
other results nor released their code, comparison with [34] is only given in the Tab.1 of our original submission. Since
we only use naive nearest neighbor based label assignment, our superiority only comes from better learned projection
function with the proposed visual structure constraints. For DCN[18], our results on AwA1,CUB,SUN (SS: **96.2, 74.2,**
**67.8, PS: 87.3, 73.4, 63.4**) are better than theirs (SS: 82.3, 55.6, 67.4, PS:65.2, 56.2, 61.8). For DIPL[A], our results on
AwA1,CUB, SUN (SS: **96.2, 74.2**, 67.8, PS: **87.3, 73.4**, 63.4, gZSL(no SUN): **81.8, 57.5**, Avg: **75.2**) are also overall
better than their results (SS: 96.1, 68.2, **70.0**, PS: 85.6, 65.4, **67.9**, gZSL: 75.6, 43.2, Avg: 71.5) except the SUN dataset.
**R2:Submission checklist e.g."error bars".** Sorry, we check it "yes" in the system by mistake and will change it.
**R3:Key differences with [34].**Though our motivation is superficially similar, the key ideas are definitely different and
complementary. ZSL methods often have two steps: projection function learning and label assignment. [34] is to
improve label assignment over naive NN using a fixed project function while we aim to learn better projection function
by only using naive NN assignment. Our better results also verified our key idea(i.e., the proposed projection learning
objective). Besides, rather than using hard matching in previous methods including [34], our WDVSc is the first to use
soft matching (Line 199-202) with probability, which brings the extra gain of WDVSc over CDVSc and BMVSc.
**R3:Key differences with [A].** [A] is an Arxiv paper and not published yet. After reading it, we find it is just a special
case (single direction version) of our CDVSc. On AwA1, CUB and SUN, its performance (88.64,58.8,86,16) is worse
than our CDVSc (89.6, 69.9, 90.6) let alone WDVSc(**92.9, 71.0, 91.2**).
**R3:Dependence on discriminative clusters and known cluster number K?** For fair comparison in the traditional
setting, our method shows that it can handle indiscriminative clusters and unknown K. These are already explained
in <span style="color:red">**Line 303 -317 and Tab.6**</span> very clearly. On fine-grained datasets CUB and SUN, we have provided their feature
distribution in the supplemental materials and the right of Tab.2 respectively. Though their clusters are not perfectly
separable, our method still achieves consistent performance gain. This is also validated on the large-scale ImageNet
dataset, which has more than 1500 unseen classes. On the right of Tab.1, the per-sample results of different K (guessed
values) are also provided. For Tab.4, it is the experiment results of the new setting where many noisy and unrelated
images are manually added. In this new setting, most existing methods leveraging unseen center priors will fail, which
also demonstrate the importance of this setting. By using the proposed simple strategy, our method works well again.
**R3:Evaluation with AUSUC.** Our AUSUC results are shown on the left of Tab.2, which are much better than the
best-reported results EXEM [C](AwA2:0.559, CUB:0.366, SUN:0.251) by a very large margin.
**R3:Quantitative evaluation of domain shift.** We have calculated the distances between the projected and the real
centers in the middle table of Tab.2. By using the proposed visual structure constraints, the distances are reduced
significantly on all the datasets including ImageNet, which indicates the domain shift problem is improved quantitatively.
**R3: New setting is ad-hoc and its evaluation.**It is indeed a very common and important setting for real industry
applications but never studied before. Quantitative evaluation is already given in Tab.4 of the original submission.

[Meta-Review · NeurIPS 2019]

The submission originally received scores mixed region that put it into the borderline region. The reviewers praised the simple and apparently effective method, but also noted a number of issues, in particular an unclear relation to [34] (which itself is rather unclear) as well as an insufficient experiment evaluation. In their response the authors provided additional information and results, which the reviewers appreciated. A detailed discussion followed, that ultimately let to the conclusion that the contribution is valuable and that authors should not be punished for a lack of clarity in the prior work [34]. Therefore, the recommendation is to accept the work. The authors are encourage to incorporate the reviewer suggestions and the material of the rebuttal into the camera-ready manuscript.